# Effective Learning with Node Perturbation in Deep Neural Networks

## Abstract

Backpropagation (BP) is the dominant and most successful method for training parameters of deep neural network models. However, BP relies on two computationally distinct phases, does not provide a satisfactory explanation of biological learning, and can be challenging to apply for training of networks with discontinuities or noisy node dynamics. By comparison, node perturbation (NP) proposes learning by the injection of noise into the network activations, and subsequent measurement of the induced loss change. NP relies on two forward (inference) passes, does not make use of network derivatives, and has been proposed as a model for learning in biological systems. However, standard NP is highly data inefficient and unstable due to its unguided noise-based search process. In this work, we investigate different formulations of NP and relate it to the concept of directional derivatives as well as combining it with a decorrelating mechanism for layer-wise inputs. We find that a closer alignment with directional derivatives together with input decorrelation at every layer significantly enhances performance of NP learning, making its performance on the train set competitive with BP and allowing its application to noisy systems in which the noise process itself is inaccessible.

## 1 Introduction

Backpropagation (BP) is the workhorse of modern artificial intelligence. It provides an efficient way of performing multilayer credit assignment, given a differentiable neural network architecture and loss function (Linnainmaa, 1970). Despite BP's successes, it requires an auto-differentiation framework for the backward assignment of credit, introducing a distinction between a forward, or inference phase, and a backward, or learning phase, increasing algorithmic complexity and impeding implementation in (neuromorphic) hardware (Kaspar et al., 2021; Zenke & Neftci, 2021). Furthermore, BP has long been criticized for its lack of biological detail and plausibility (Grossberg, 1987; Crick, 1989; Lillicrap et al., 2016), with significant concerns again being the two separate forward and backward phases, but also its reliance on gradient propagation and the non-locality of the information required for credit assignment.

Alternative algorithms have been put forth over the years, though their inability to scale and difficulty in achieving levels of performance comparable to BP have held back their use. One such algorithm is node perturbation (NP) (Dembo & Kailath, 1990; Cauwenberghs, 1992). In NP, node activations are perturbed by a small amount of random noise. Weights are then updated to produce the perturbed activations in proportion to the degree by which the perturbation improved performance. This method requires a measure of the loss function being optimised on a network both before and after the inclusion of noise. Such an approach is appealing because it leverages the same forward pass twice, rather than relying on two computationally distinct phases. It also does not require non-local information other than the global performance signal.

Despite these benefits, Hiratani et al. (2022) demonstrate that NP is extremely inefficient compared to BP, requiring two to three orders of magnitude more training cycles, depending on network depth and width. In addition, they found training with NP to be unstable, in many cases due to exploding weights. Another phenomenon uncovered in their work is that the covariance of NP's updates is significantly higher than that of BP, which can mathematically be described as an effect mediated by correlations in the input data.

Another proposed approach, referred to as weight propagation (WP), is to inject noise directly into the weights of a network, rather than the activations (Werfel et al., 2003; Fiete & Seung, 2006). This method shows similarities to evolutionary algorithms and can outperform NP in special cases (Züge et al., 2021). A downside to WP is that there are many more weights than nodes in neural networks, making the exploration space larger thus slowing down learning. In fact, both NP and WP lag far behind BP in terms of efficiency. This is to be expected, however, as noise perturbations conduct a random search through parameter space, rather than being gradient directed.

In this work, we put forward three contributions. First, we reframe the process of node perturbation together with the subsequent measurement of the output loss change in terms of directional derivatives. This provides a more solid theoretical foundation for node perturbation and, consequently, a different update rule, which we refer to as iterative node perturbation. Directional derivatives have been related to NP before in the context of forward gradient learning (Baydin et al., 2022; Ren et al., 2022). This approach is different from ours in that the perturbations are used to estimate gradients, but perturbed forward passes are not actually performed, making these approaches unsuitable for use in noisy systems.

Second, we introduce an approximation to iterative node perturbation, referred to as activity-based node perturbation, which is more efficient and has the additional advantage that it can be implemented in noisy systems such as imprecise hardware implementations (Gokmen, 2021) or biological systems (Faisal et al., 2008), where the noise itself is not measurable.

Third, we propose to use a decorrelation method, first described by Ahmad et al. (2023), to debias layer-wise activations and thereby achieve faster learning. Because NP-style methods directly correlate perturbations of unit activities with changes in a reward signal, decorrelation of these unit activities helps to eliminate confounding effects, making credit assignment more straightforward. In addition, as demonstrated by Hiratani et al. (2022), correlations in the input data lead to more bias in NP's updates by increasing their covariance. By combining decorrelation with the different NP methods, we find that it is possible to achieve orders of magnitude increase in model convergence speed, with performance levels rivalling networks trained by BP in certain contexts.

## 2  METHODS

### 2.1  NODE PERTURBATION AND ITS FORMULATIONS

Let us define the forward pass of a fully-connected neural network, with $L$ layers, such that the output of a given layer, $l \in 1, 2, \ldots, L$ is given by

$$\mathbf{x}_l = f\left(\mathbf{a}_l\right),$$

where $\mathbf{a}_l = \mathbf{W}_l \mathbf{x}_{l-1}$ is the pre-activation with weight matrix $\mathbf{W}_l$, $f$ is the activation function and $\mathbf{x}_l$ is the output from layer $l$. The input to our network is therefore denoted $\mathbf{x}_0$, and the output $\mathbf{x}_L$. We consider learning rules which update the weights of such a network in the form

$$\mathbf{W}_l \leftarrow \mathbf{W}_l + \eta_W \Delta \mathbf{W}_l,$$

where $\eta_W$ is a small, constant learning rate, and $\Delta \mathbf{W}_l$ is a parameter update direction derived from a given algorithm (BP, NP, etc.). Recall that the regular BP update is given by

$$\Delta \mathbf{W}_l^{\text{BP}} = -\mathbf{g}_l \mathbf{x}_{l-1}^\top \tag{1}$$

with $\mathbf{g}_l = \frac{\partial \mathcal{L}}{\partial \mathbf{a}_l}$ the gradient of the loss $\mathcal{L}$ with respect to the layer activations $\mathbf{a}_l$. In the following, we consider weight updates relative to one pair of inputs $\mathbf{x}_0$ and targets $\mathbf{t}$. In practice, these updates are averaged over mini-batches.

#### 2.1.1  TRADITIONAL NODE PERTURBATION

In the most common formulation of NP, noise is injected into each layer's pre-activations and weights are updated in the direction of the noise if the loss improves and in the opposite direction if it worsens. Two forward passes are required: one clean and one noise-perturbed. During the noisy pass, noise is injected into the pre-activation of each layer to yield a perturbed output

$$\tilde{\mathbf{x}}_l = f\left(\tilde{\mathbf{a}}_l + \boldsymbol{\epsilon}_l\right) = f\left(\mathbf{W}_l \tilde{\mathbf{x}}_{l-1} + \boldsymbol{\epsilon}_l\right), \tag{2}$$

where the added noise $\boldsymbol{\epsilon}_l \sim \mathcal{N}(\mathbf{0}, \sigma^2 \mathbf{I}_l)$ is a spherical Gaussian perturbation with no cross-correlation and $\mathbf{I}_l$ is an $N_l \times N_l$ identity matrix with $N_l$ the number of nodes in layer $l$. Note that this perturbation has a non-linear effect as the layer's perturbed output $\tilde{\mathbf{x}}_l$ is propagated forward through the network, resulting in layers deeper in the network being perturbed slightly more than earlier layers.

Having defined a clean and noise-perturbed network-pass, we can measure a loss differential for a NP-based update. Supposing that the loss $\mathcal{L}$ is measured using the network outputs, the loss difference between the clean and noisy network is given by

$$\delta \mathcal{L} = \mathcal{L}(\tilde{\mathbf{x}}_L) - \mathcal{L}(\mathbf{x}_L) \,,$$

where $\delta \mathcal{L}$ is a scalar measure of the difference in loss induced by the addition of noise to the network. Given this loss difference and the network's perturbed and unperturbed outputs, we compute a layer-wise learning signal:

$$\Delta \mathbf{W}_l^{\mathrm{NP}} = -\frac{\delta \mathcal{L}}{\sigma} \boldsymbol{\epsilon}_l \mathbf{x}_{l-1}^\top \,. \tag{3}$$

### 2.1.2 Iterative node perturbation

Though mathematically simple, the traditional NP approach described above has a number of drawbacks. One of the biggest drawbacks is that all layers are updated simultaneously and each layer's noise impact is confounded by additional noise in previous and following layers. Appendix A describes how correlations arise between layers. This creates a mismatch between traditional NP and a theoretically grounded approach for derivative measurement.

In the following, we develop a more principled approach to node perturbation. Our goal is to approximate the gradient of the loss with respect to the pre-activations in a layer $l$ without needing to propagate gradients. To this end, we consider the partial derivative of the loss with respect to the pre-activation $a_l^i$ of unit $i$ in layer $l$ for all $i$. We define a perturbed state as

$$\tilde{\mathbf{x}}_k(h) = f\left(\tilde{\mathbf{a}}_k + h\mathbf{m}_k\right)$$

with $h$ an arbitrary scalar and binary vectors $\mathbf{m}_k = \mathbf{e}_i$ if $k = l$ with $\mathbf{e}_i$ a standard unit vector and $\mathbf{m}_k = \mathbf{0}$ otherwise. We may now define the partial derivatives as

$$(\mathbf{g}_l)_i = \lim_{h \to 0} \frac{\mathcal{L}(\tilde{\mathbf{x}}_L(h)) - \mathcal{L}(\mathbf{x}_L)}{h} \,.$$

This suggests that node perturbation can be rigorously implemented by measuring derivatives using perturbations $h\mathbf{m}_k$ for all units $i$ individually in each layer $l$. However, this would require as many forward-passes as there exist nodes in the network, which would be extremely inefficient.

An alternative approach is to define perturbations in terms of directional derivatives. Directional derivatives measure the derivative of a function based upon an arbitrary vector direction in its dependent variables. However, this can only be accomplished for a set of dependent variables which are individually independent of one-another. Thus, we cannot compute such a measure across an entire network. We can, however, measure the directional derivative with respect to a specific layer via a perturbation given by

$$\tilde{\mathbf{x}}_k(h) = f\left(\tilde{\mathbf{a}}_k + h\mathbf{v}_k\right) \,,$$

where $\mathbf{v}_k \sim \mathcal{N}(0, \mathbf{I}_k)$ if $k = l$ and $\mathbf{v}_k = \mathbf{0}$ otherwise. Given this definition, we can directly measure a directional derivative of our deep neural network for layer $l$ in vector direction $\mathbf{v} = (\mathbf{v}_1, \dots, \mathbf{v}_L)$ as

$$\nabla_{\mathbf{v}} \mathcal{L} = \lim_{h \to 0} \frac{\mathcal{L}\left(\tilde{\mathbf{x}}_L(h)\right) - \mathcal{L}\left(\mathbf{x}_L\right)}{h \|\mathbf{v}\|_F} \,.$$

By sampling the noise vector repeatedly, we arrive at the gradient of the loss with respect to a specific layer:

$$\mathbf{g}_l \approx \sqrt{N_l} \langle \nabla_{\mathbf{v}} \mathcal{L} \mathbf{v} \rangle_{\mathbf{v}} \,,$$

which normalizes the length of the measured gradient vector based upon the projection length of the unit-vectors sampled for the directional derivative onto the parameter axes.

This description allows us to accurately measure the gradient of a particular layer of a deep neural network by perturbation. This can remain rather inefficient, given that the number of perturbation

measures is now proportional to the number of layers in the network, however, it is much more efficient than individual node perturbations. We refer to this method, with weight update

$$\Delta \mathbf{W}_l^{\text{INP}} = -\sqrt{N_l} \, \langle \nabla_{\mathbf{v}} \mathcal{L} \mathbf{v} \rangle_{\mathbf{v}} \, \mathbf{x}_{l-1}^{\top} \qquad (4)$$

as iterative node perturbation (INP).

### 2.1.3 ACTIVITY-BASED NODE PERTURBATION

Though theoretically well grounded, the iterative update method described above incurs significant computational overhead compared to the direct noise-based update method. Specifically, in order to precisely compute directional derivatives, noise must be applied in a purely layer-wise fashion, requiring a high degree of control over the noise injection. This results in a less biologically plausible and functionally more challenging implementation with less potential for scaling.

Stepping beyond this method, we attempt to balance learning speed and computational cost by approximating this directional derivative across the whole network simultaneously. This involves assuming that all node activations in our network are independent and treating the entire network as if it were a single layer. This can be achieved by, instead of measuring and tracking the injected noise alone, measuring the state difference between the clean and noisy forward passes of the whole network. Concretely, taking the definition of the forward pass given by NP in Eq. 2, we define

$$\Delta \mathbf{W}_l^{\text{ANP}} = -\sqrt{N} \, \delta \mathcal{L} \, \frac{\delta \mathbf{a}_l}{||\delta \mathbf{a}||_F} \, \mathbf{x}_{l-1}^{\top} \, , \qquad (5)$$

where $N = \sum_{l=0}^{L} N_l$ is the total number of units in the network, $\delta \mathbf{a}_l = \tilde{\mathbf{a}}_l - \mathbf{a}_l$ is the activity difference between a noisy and clean pass in layer $l$ and $\delta \mathbf{a} = (\delta \mathbf{a}_1, \ldots, \delta \mathbf{a}_L)$ is the concatenation of all activity differences. We refer to this update rule as activity-based node perturbation (ANP).

Here, we have now updated multiple aspects of the NP rule. First, rather than using the measure of noise injected at each layer, we instead measure the total change in activation between the clean and noisy networks. This aligns our measure more closely with the directional derivative: we truly measure how the network changed in its response rather than ignoring the impact of all previous layers and their noise upon the current layer.

This direct use of the activity difference also requires a recomputation of the scale of the perturbation vector. Here we carry the rescaling out by a normalization based upon the activity-difference length and then upscale the signal of this perturbation based upon the network size.

Note that our directional derivative measuring process remains theoretically well grounded, but is now potentially biased in terms of the vector directions it measures derivatives in. This is due to biasing induced by noise propagated from previous layers with a correlated impact upon activity differences.

### 2.2 INCREASING NP EFFICIENCY THROUGH DECORRELATION

Uncorrelated data variables have been proposed and demonstrated as impactful in making credit assignment more efficient in deep neural networks (LeCun et al., 2002). If a layer's inputs, $\mathbf{x}_l$, have highly correlated features, a change in one feature can be associated with a change in another correlated feature, making it difficult for the network to disentangle the contributions of each feature to the loss. This can lead to less efficient learning, as has been described in previous research in the context of BP (Luo, 2017; Wadia et al., 2021). NP additionally benefits from decorrelation of input variables at every layer. Specifically, Hiratani et al. (2022) demonstrate that the covariance of NP updates between layers $k$ and $l$ can be described as

$$C_{kl}^{\text{np}} \approx 2 C_{kl}^{\text{sgd}} + \delta_{kl} \left\langle \sum_{m=1}^{k} ||\mathbf{g}_m||^2 \mathbf{I}_k \otimes \mathbf{x}_{k-1} \mathbf{x}_{k-1}^T \right\rangle_{\mathbf{x}} \, ,$$

where $C_{kl}^{\text{sgd}}$ is the covariance of SGD updates, $\delta_{kl}$ is the Kronecker delta and $\otimes$ is a tensor product. The above equation implies that in NP the update covariance is twice that of the SGD updates plus an additional term that depends on the correlations in the input data, $\mathbf{x}_{k-1} \mathbf{x}_{k-1}^T$. Removing correlations

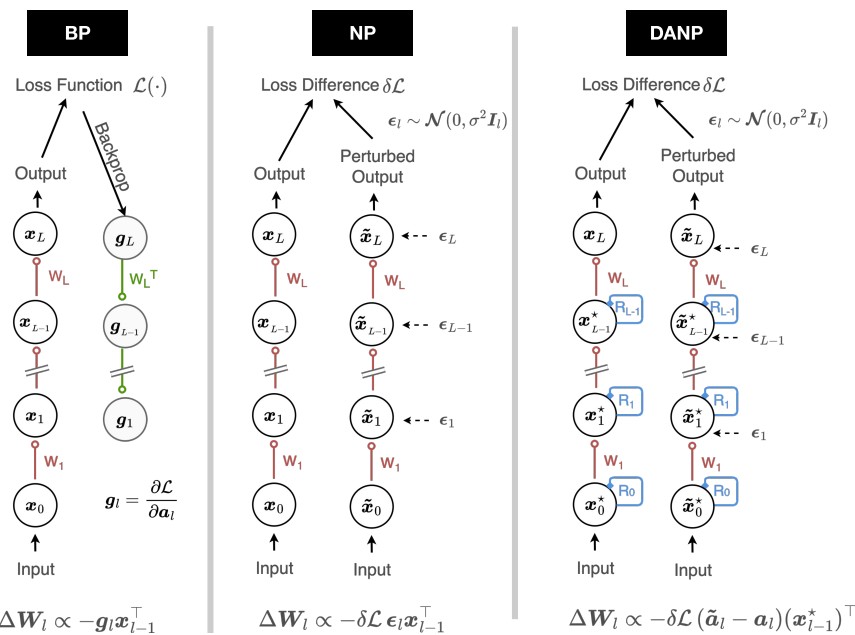

Figure 1: A graphical illustration of the computations required for update measurement for backprop-agation (left), node perturbation (middle), and decorrelated activity-based node perturbation (right).

from the input data should therefore reduce the bias in the NP algorithm updates, possibly leading to better performance.

In this work, we introduce decorrelated node perturbation, in which we decorrelate each layer's input activities using a trainable decorrelation procedure first described by Ahmad et al. (2023). A layer input $\mathbf{x}_l$ is decorrelated by multiplication by a decorrelation matrix $\mathbf{R}_l$ to yield a decorrelated input $\mathbf{x}_l^\star = \mathbf{R}_l \mathbf{x}_l$. The decorrelation matrix $\mathbf{R}_l$ is then updated according to

$$\mathbf{R}_l \leftarrow \mathbf{R}_l - \eta_R \left( \mathbf{x}_l^\star \left( \mathbf{x}_l^\star \right)^\top - \mathrm{diag}\left( \left( \mathbf{x}_l^\star \right)^2 \right) \right) \mathbf{R}_l \,,$$

where $\eta_R$ is a small constant learning rate and $\mathbf{R}_l$ is initialised as the identity matrix. For a full derivation of this procedure see (Ahmad et al., 2023).

Decorrelation can be combined with any of the formulations described in the previous paragraphs, which we refer to as DNP, DINP and DANP for regular, iterative and activity-based node perturbation, respectively. In Appendix B, Algorithm 1, we describe decorrelation using the activity-based update procedure. In Figure 1 we illustrate the operations carried out to compute updates for some of the methods explained above.

## 2.3 EXPERIMENTAL VALIDATION

To measure the performance of the algorithms proposed, we ran a set of experiments with the CIFAR-10 dataset (Krizhevsky, 2009), specifically aiming to quantify the performance differences between the traditional (NP), layer-wise iterative (INP) and activity-based (ANP) formulations of NP. These experiments were run using a three-hidden-layer fully-connected neural network with leaky ReLU activation functions and a mean squared error (MSE) loss on the one-hot encoding of class membership. Experiments were repeated using five different random seeds, after which performance statistics were averaged. The learning rates were determined for each algorithm separately by a grid search, in which the learning rate started at $10^{-6}$ and was doubled until performance stopped increasing. See Appendix C for the learning rates used in the experiments. When using NP methods, activity perturbations were drawn from a univariate Gaussian with variance $\sigma^2 = 10^{-6}$.

Another experiment was run with CIFAR-10 to determine the performance effects of decorrelation. This was done in both a single-layer and a three-hidden-layer network, so that the scaling properties

of the algorithms could be studied. Decorrelated BP (DBP) was also added as baselines for this and subsequent simulations. The specific architecture constructions are described in Appendix D.

Additionally, two scaling experiments were run. One using CIFAR-10 with 3, 6, or 9 fully-connected hidden layers and another one using CIFAR-100 and a convolutional architecture with 3 convolutional layers, followed by a fully-connected hidden layer and an output layer. The purpose of these experiments was to assess whether the algorithms scale to deeper architectures and higher output dimensions. The latter has been shown to be a specific issue for the traditional NP algorithm (Hiratani et al., 2022).

Finally, a version of the DANP algorithm was run where two noisy forward passes were used, instead of a clean and a noisy pass. This experiment specifically investigates whether DANP might work in inherently noisy systems.

## 3   RESULTS

### 3.1   COMPARING NP FORMULATIONS

In single-layer networks, the three described NP formulations (NP, INP and ANP) converge to an equivalent learning rule. Therefore, multi-layer networks with three hidden layers were used to investigate performance differences across our proposed formulations.

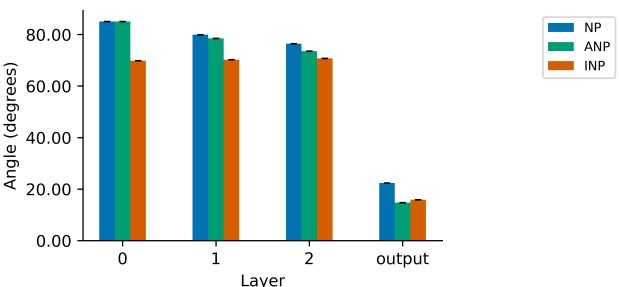

Figure 2: Angles of various NP method updates with respect to BP updates, collected by averaging over 100 different noise samples during training.

Figure 2 shows how weight updates of the different node perturbation methods compare to those from BP in a three-hidden-layer, fully-connected, feedforward networks for the CIFAR-10 classification task. When measuring the angles of comparison between the various methods, we can observe that the INP method is by far the most well-aligned in its updates with respect to backpropagation, closely followed by ANP.

These results align with the theory laid out in the methods section of this work. Note that for all of these algorithms, alignment with BP updates improve when updates are averaged over more samples of noise. Therefore, it is the ranking of the angles between the NP algorithms that is of interest here, not the absolute value of the angle itself, as all angles would improve with more noise samples.

### 3.2   IMPACT OF DECORRELATION

To assess the impact of decorrelation on NP's performance, we studied a single-layer network trained with NP, DNP, BP and DBP. Note that the different formulations of NP are identical in a single-layer networks where there is no impact on activities in the layer from past layers.

Figure 3 shows that, when training single-layer networks, NP holds up relatively well compared to BP. This is likely due to the limited number of output nodes that need to be perturbed, making NP a decent approximation of BP in this regime. By comparison, DNP converges *faster* than both BP and NP, despite using only a global reward signal. DNP's benefit is less pronounced in test accuracy compared to train accuracy. As in (Ahmad et al., 2023), the benefit of decorrelation for learning was also observed in decorrelated backpropagation (DBP).

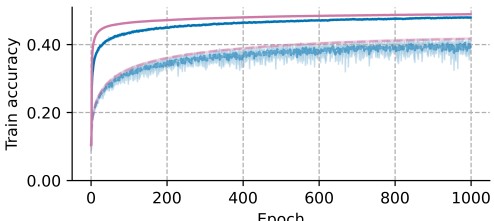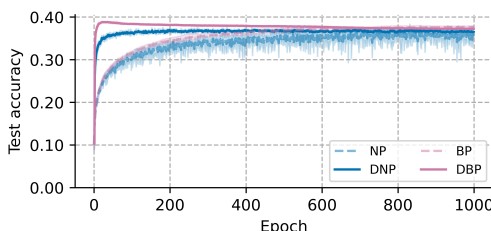

Figure 3: Performance of node perturbation and backpropagation with and without decorrelation on CIFAR-10 when training fully-connected single-layer architecture. Note that all NP methods have an equivalent formulation in a single-layer network.

It appears that part of the benefit of decorrelation is due to a lack of correlation in the input unit features and a corresponding ease in credit assignment without confound. An additional benefit from decorrelation, that is specific to NP-style updates, is explained by the way in which decorrelation reduces the covariance of NP weight updates, as described in the methods section above.

### 3.3 NEURAL NETWORK PERFORMANCE COMPARISONS

We proceed by determining the performance of the different algorithms in multi-layer neural networks. See Appendix E for peak accuracies.

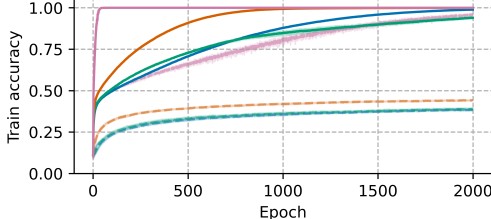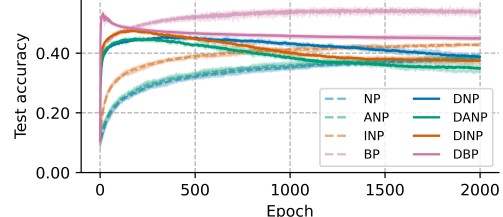

Figure 4: Performance of different NP and BP formulations on CIFAR-10 when training fully-connected three-hidden-layer architectures.

Figure 4, shows the various network performance measures when training a multi-layer fully connected network. Though the measured angles during training differ between these algorithms, we can see that this does not appear to have a significant effect in distinguishing the ANP and NP methods. INP does show a performance benefit relative to the other two formulations. Also, all versions of DNP are competitive with BP in terms of train accuracy, with DINP even outperforming BP. For test accuracy, BP remains the best algorithm and DBP performs by far the best on the train set. Note that DNP performs much better in a three-hidden-layer network than in the single-layer networks explored in Figure 3, meaning that DNP does facilitate multi-layer credit assignment much more than regular NP.

Figure 5 compares the performance of DANP and DINP for 3, 6 and 9 hidden layers. Performance for both algorithms is robust across network depth, with DINP outperforming DANP at every network depth. Though increasing the number of hidden layers beyond three did not boost performance, it also did not make the algorithms unstable, showing potential for the use of DNP in deeper networks. This is an area for future research.

To further investigate the scaling and generalization capacities of our approach, we also trained and tested convolutional architectures for image classification. Figure 6 shows that, when training a convolutional neural network on CIFAR-100, DANP massively outperforms ANP in both train and test accuracy, with DINP performing even better. Both DNP formulations outperform BP on the train set and DINP also outperforms BP on the test set. DBP also outperforms BP, showing the best test

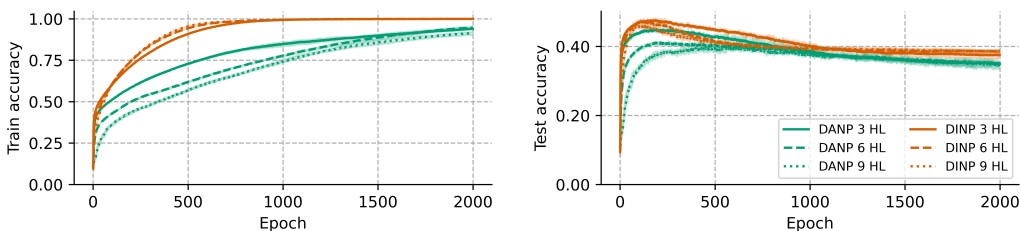

Figure 5: Performance of DANP and DINP for 3-, 6- and 9-hidden-layer networks.

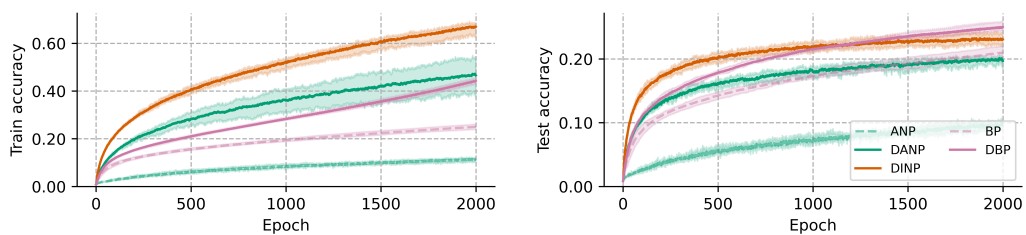

Figure 6: Performance of different node perturbation and backpropagation variants when training a convolutional neural network on CIFAR-100.

performance of all algorithms and catching up with DANP, but not DINP, late in training in terms of train performance.

Note that performance of all algorithms is quite low compared to CIFAR-100 benchmarks as we are using a relatively shallow convolutional neural network in combination with MSE loss, which is not the standard for classification problems with a large output space. The purpose of this experiment was not to attain competitive classification performance per-se, but a comparative study of BP and DNP under a simple training regime. These results also illustrate that the application of such perturbation-based methods can extend trivially beyond fully-connected architectures.

### 3.4 NODE PERTURBATION FOR NOISY SYSTEMS

One of the most interesting applications of perturbation-based methods for learning are for systems which cannot operate without the possibility of noise. This includes both biological nervous systems as well as a range of neuromorphic hardware architectures. In order to demonstrate that this method is also applicable to architectures with embedded noise, we train networks in which there is no clean network pass available. Instead, two noisy network passes are computed and one is taken as if it were the clean pass. That is, we use $\delta \mathbf{a}_l = \tilde{\mathbf{a}}_l^{(1)} - \tilde{\mathbf{a}}_l^{(2)}$ in Eq. 5. This is similar in spirit to the approach suggested by Cho et al. (2011).

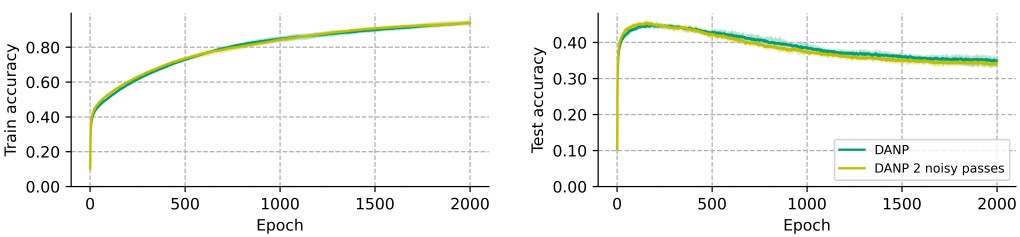

Figure 7: Performance of DANP when applied with one clean network and one noisy network, vs with two noisy network passes. These curves are for a three-layer fully-connected network trained for CIFAR-10 classification (comparable to Figure 4).

In this case we specifically compare and apply (decorrelated) activity-based node perturbation. This method does not assume that the learning algorithm can independently measure noise. Instead it can only measure the present, and potentially noisy, activity and thereafter measure activity differences to direct learning.

As can be seen in Figure 7, computing updates based upon a set of two noisy passes, rather than a clean and noisy pass, produces extremely similar learning dynamics with some minimal loss in performance. The similarity of the speed of learning and performance levels suggests that clean network passes may provide little additional benefit for the computation of updates in this regime.

## 4  DISCUSSION

In this work, we explored several formulations of NP learning and also introduced a layer-wise decorrelation method which significantly outperforms the baseline implementations. We attribute this increased efficacy to a more efficient credit assignment by virtue of decorrelated input features at every layer, as well as an attenuation of the bias in NP's updates caused by their covariance.

In our results we see robust speedups in training across architectures. This speedup is sufficient to suggest that such alternative formulations of NP could prove sufficiently fast as to be competitive with traditional BP in certain contexts. The inclusion of decorrelation does appear to have an impact on the level of overfitting in these systems, with our NP-trained networks having a generally lower test accuracy. This fits with recent research which suggests an overfitting impact of data whitening (Wadia et al., 2021). In this respect, further investigation is warranted.

Exploring more efficient forms of noise-based learning is interesting beyond credit-assignment alone. First, this form of learning is more biologically plausible as it does not require weight transport of any kind, or even any specific feedback connectivity or computations. There is ample evidence for noise in biological neural networks and we suggest here that this could be effectively used for learning. Decorrelation in the brain is less well evidenced, however various mechanisms act to reduce correlation or induce whitening - especially in early visual cortical areas (King et al., 2013). Additionally, lateral inhibition, which is known to occur in the brain, can be interpreted as a way to reduce redundancy in input signals akin to decorrelation, making outputs of neurons less similar to each other (Békésy, 1967).

In addition, noise-based learning approaches might be more efficiently implemented in hardware. First, as demonstrated in Figure 7, our proposed DANP algorithm scales well even when there is no access to a 'clean' model pass. This means that such an approach could be ideally suited for application in noisy inference hardware. Even on traditional hardware architectures, forward passes are often easier to optimize than backward passes and are often significantly faster to compute. This can be especially true for neuromorphic computing approaches, where backward passes require automatic differentiation implementations and a separate computational pipeline for backward passes. In both of these cases, a noise-based approach to learning could prove highly efficient.

Though the presented methods for effective noise-based learning show a great deal of promise, there are a number of additional research steps to be taken. The architectures considered are relatively shallow, and thus an investigation into how well this approach scales for very deep networks would be beneficial. Testing the scalability of these approaches to tasks of greater complexity is also crucial, as are their application to other network architectures such as residual networks and transformers.

In general, our work opens up exciting opportunities since it has the potential to bring gradient-free training of deep neural networks within reach. That is, in addition to not requiring a backward pass, efficient noise-based learning may also lend itself to networks not easily trained by backpropagation, such as those consisting of activation functions with jumps, binary networks or networks in which the computational graph is broken, as in reinforcement learning.

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

## A    NOISE PROPAGATES WEIGHT CORRELATIONS

Traditional NP updates have a number of sources of error. One reason is that use of the injected noise $\epsilon_l$ to compute the learning signal ignores how the output activity of layer $l$ is not only impacted by the noise injected directly into it, but also by the cumulative effect of perturbations added to previous layers. If the noise is randomly sampled from a Gaussian distribution with mean zero, one might be tempted to simply assume that the perturbations from previous layers cancel out in expectation, but this assumption ignores the correlations introduced into these perturbations by the network's weight matrices at all preceding layers. Multiplying a random vector by a non-orthogonal matrix will introduce a non-random covariance structure into it. To see why, consider a transformation

$$\mathbf{W}\mathbf{x} = \mathbf{y}\,, \tag{6}$$

where $W$ is a randomly initialized weight matrix and $x$ is an uncorrelated input vector for which the following holds in expectation:

$$\langle \mathbf{x}\mathbf{x}^\top \rangle = \mathbf{I} \tag{7}$$

with $\mathbf{I}$ the identity matrix. The covariance matrix of $\mathbf{y}$ can be described as

$$
\begin{aligned}
\mathrm{cov_y} &= \langle \mathbf{y}\mathbf{y}^\top \rangle \\
&= \langle (\mathbf{W}\mathbf{x})(\mathbf{W}\mathbf{x})^T \rangle \\
&= \langle (\mathbf{W}\mathbf{x})(\mathbf{x}^\top \mathbf{W}^\top) \rangle \\
&= \langle \mathbf{W}\mathbf{x}\mathbf{x}^\top \mathbf{W}^\top \rangle \\
&= \mathbf{W}\langle \mathbf{x}\mathbf{x}^\top \rangle \mathbf{W}^\top \\
&= \mathbf{W}\mathbf{I}\mathbf{W}^\top \\
&= \mathbf{W}\mathbf{W}^\top
\end{aligned}
$$

Therefore, any matrix for which $\mathbf{W}\mathbf{W}^\top \neq \mathbf{I}$ will add some covariance structure into output vector $\mathbf{y}$, even when input vector $\mathbf{x}$ is uncorrelated.

# B ALGORITHM PSEUDOCODE

---

**Algorithm 1** Decorrelated activity-based node perturbation (DANP).

---

**Input:** *data* $\mathcal{D}$, *network* $\{(\mathbf{W}_l, \mathbf{R}_l)\}_{l=1}^{L}$, *learning rates* $\eta_W$ *and* $\eta_R$

**for each** *epoch* **do**

    **for each** $(\mathbf{x_0}, \mathbf{t}) \in \mathcal{D}$ **do**

        **for** layer $l$ **from** 1 **to** $L$ **do**                                        $\triangleright$ Regular forward pass

             $\mathbf{x}_{l-1}^{\star} = \mathbf{R}_{l-1}\mathbf{x}_{l-1}$

             $\mathbf{a}_l = \mathbf{W}_l \mathbf{x}_{l-1}^{\star}$

             $\mathbf{x}_l = f(\mathbf{a}_l)$

        **end for**

         $\tilde{\mathbf{x}}_0 = \mathbf{x}_0$

        **for** layer $l$ **from** 1 **to** $L$ **do**                                            $\triangleright$ Noisy forward pass

             $\tilde{\mathbf{x}}_{l-1}^{\star} = \mathbf{R}_{l-1}\tilde{\mathbf{x}}_{l-1}$

             $\tilde{\mathbf{a}}_l = \mathbf{W}_l \tilde{\mathbf{x}}_{l-1}^{\star} + \boldsymbol{\epsilon}_l$

             $\tilde{\mathbf{x}}_l = f(\tilde{\mathbf{a}}_l)$

        **end for**

         $\delta\mathcal{L} = (\mathbf{t} - \tilde{\mathbf{x}}_L)^2 - (\mathbf{t} - \mathbf{x}_L)^2$                         $\triangleright$ Compute loss difference

        **for** layer $l$ **from** 1 **to** $L$ **do**

             $\mathbf{W}_l \leftarrow \mathbf{W}_l - \eta_W \sqrt{N} \, \delta\mathcal{L} \, \frac{\tilde{\mathbf{a}}_l - \mathbf{a}_l}{||\delta\mathbf{a}||_F} \left(\mathbf{x}_l^{\star}\right)^{\top}$        $\triangleright$ Update weight matrix

             $\mathbf{R}_l \leftarrow \mathbf{R}_l - \eta_R \left(\mathbf{x}_l^{\star} \left(\mathbf{x}_l^{\star}\right)^{\top} - \text{diag}\left(\left(\mathbf{x}_l^{\star}\right)^2\right)\right) \mathbf{R}_l$     $\triangleright$ Update decorrelation matrix

        **end for**

    **end for**

**end for**

---

## C  LEARNING RATES

Table 1 shows the learning rates $\eta_W$ used in each experiment. For each experiment, the highest stable learning rate was selected. The search over learning rates started at $10^{-6}$ and the learning rate was doubled until the highest stable learning rate was found. For the decorrelation learning rate a fixed value of $\eta_R = 10^{-3}$ was chosen based on a prior manual exploration.

Table 1: Learning rates used for different learning algorithms and architectures.

| METHOD | 1 LAYER | 3 HIDDEN LAYERS | 6 HIDDEN LAYERS | 9 HIDDEN LAYERS | CONVNET |
|---|---|---|---|---|---|
| NP | 8.2 | $2.6 \times 10^{-1}$ | – | – | – |
| DNP | $1.0 \times 10^3$ | $1.3 \times 10^2$ | – | – | – |
| ANP | – | $2.6 \times 10^{-1}$ | – | – | 16 |
| DANP | – | $1.3 \times 10^2$ | 66 | 66 | $5.2 \times 10^2$ |
| INP | – | 1.0 | – | – | – |
| DINP | – | $2.6 \times 10^2$ | $2.6 \times 10^2$ | $2.6 \times 10^2$ | $2.1 \times 10^3$ |
| BP | $8.0 \times 10^{-6}$ | $6.4 \times 10^{-5}$ | – | – | $5.1 \times 10^{-4}$ |
| DBP | $1.0 \times 10^{-3}$ | $4.1 \times 10^{-3}$ | – | – | $1.0 \times 10^{-3}$ |

# D  NEURAL NETWORK ARCHITECTURES

Table 2: Details of the employed neural network architectures.

| Network | Number of Layers | Layer Types | Layer Details |
|---|---|---|---|
| Single layer | 1 | FC | 10 |
| Three hidden layers | 4 | FC | 1024 |
| | | FC | 1024 |
| | | FC | 1024 |
| | | FC | 10 |
| Six hidden layers | 7 | FC | 1024 |
| | | FC | 1024 |
| | | FC | 1024 |
| | | FC | 1024 |
| | | FC | 1024 |
| | | FC | 1024 |
| | | FC | 10 |
| Nine hidden layers | 10 | FC | 1024 |
| | | FC | 1024 |
| | | FC | 1024 |
| | | FC | 1024 |
| | | FC | 1024 |
| | | FC | 1024 |
| | | FC | 1024 |
| | | FC | 1024 |
| | | FC | 1024 |
| | | FC | 10 |
| ConvNet | 5 | Conv | $3\times3\times16, 2$ |
| | | Conv | $3\times3\times32, 2$ |
| | | Conv | $3\times3\times64, 1$ |
| | | FC | 1024 |
| | | FC | 100 |

# E    PEAK ACCURACIES

Table 3: Peak percentage accuracies for different learning algorithms and architectures.

| METHOD | 1 LAYER | | 3 HIDDEN LAYERS | | 6 HIDDEN LAYERS | | 9 HIDDEN LAYERS | | CONVNET | |
|---|---|---|---|---|---|---|---|---|---|---|
| | TRAIN | TEST | TRAIN | TEST | TRAIN | TEST | TRAIN | TEST | TRAIN | TEST |
| NP | 41.6 | 38.4 | 39.2 | 39.2 | – | – | – | – | – | – |
| DNP | 48.2 | 37.5 | 99.0 | 46.2 | – | – | – | – | – | – |
| ANP | – | – | 39.6 | 39.7 | – | – | – | – | 12.4 | 10.7 |
| DANP | – | – | 94.6 | 45.7 | 95.5 | 41.6 | 92.1 | 40.6 | 54.9 | 20.7 |
| INP | – | – | 44.5 | 43.5 | – | – | – | – | – | – |
| DINP | – | – | **100.0** | 48.2 | **100.0** | **47.9** | **100.0** | **46.8** | **69.2** | 24.4 |
| BP | 42.0 | 38.0 | 96.7 | **55.7** | – | – | – | – | 26.1 | 22.0 |
| DBP | **49.0** | **39.0** | **100.0** | 54.1 | – | – | – | – | 45.6 | **25.9** |

