# OpenReview forum: "Effective Learning by Node Perturbation in Deep Neural Networks"
_ICLR.cc/2024/Conference — Submitted to ICLR 2024_

### Official Review · Reviewer_PZS3 · 2023-10-27

**Soundness:** 2 fair
**Presentation:** 3 good
**Contribution:** 1 poor
**Rating:** 5
**Confidence:** 4

**Summary:**

The authors combine the node perturbation (NP) algorithm with an activity decorrelation method and better gradient estimation to improve NP's performance.

**Strengths:**

1. The paper does a good job introducing note perturbation (NP), along with its motivation and caveates.
2. The combination of the decorrelation method and NP is novel and speeds up training in some cases.
3. These ideas might be applied in other settings where learning is easier; e.g. weight perturbation was applied to finetuning LLMs https://arxiv.org/pdf/2305.17333.pdf

**Weaknesses:**

My main issue with the paper is that NP still performs very poorly, even with decorrelation (NP-D), and doesn't match backprop. The results do not justify the last line of the abstract, which says
> significantly enhances performance of NP learning making it competitive with BP.

Fig. 3 shows that with early stopping NP-D can slightly outperform NP, but the gap between the best performance for those two methods is smaller than the gap between BP and NP-D. Fig. 5 looks a bit more favourable, but NP-D still performs much worse than BP. Both figures show experiments on shallow networks, so it's reasonable to expect the gap will widen for deeper networks due to NP incorrectly estimating gradients.

In addition, both Fig. 3 and Fig. 5 show that NP-D results in a very large, much larger than for ND, generalization gap.

The experiments are also rather limited: it's only two very similar datasets, and only shallow architectures. Convnets are also not used for cifar10 -- a few conv layers trained with BP can perform much better than ~56% with small MLPs, so seeing how NP-D performs there would be helpful.

**Questions:**

Why is everything trained for so many epochs? 2000 epochs is a lot, and backprop should normally converge to very good performance on cifar10 within the first few dozen epochs. But here it takes hundreds.

Beginning of page 2:  weight propagation (WP) -> perturbation?

---

> ### Author Response · Authors · 2023-11-23
>
> We thank the reviewer for their thoughtful review and commentary.
>
> - My main issue with the paper is that NP still performs very poorly, even with decorrelation (NP-D), and doesn't match backprop. The results do not justify the last line of the abstract, which says
> significantly enhances performance of NP learning making it competitive with BP.
> Fig. 3 shows that with early stopping NP-D can slightly outperform NP, but the gap between the best performance for those two methods is smaller than the gap between BP and NP-D. Fig. 5 looks a bit more favourable, but NP-D still performs much worse than BP. Both figures show experiments on shallow networks, so it's reasonable to expect the gap will widen for deeper networks due to NP incorrectly estimating gradients.
> In addition, both Fig. 3 and Fig. 5 show that NP-D results in a very large, much larger than for ND, generalization gap.
>
> The abstract and other relevant passages have been updated to reflect this competitiveness is limited to certain contexts. We will also describe the overfitting problem a bit more. NP-D performs really well in terms of training accuracy but does suffer from lower test accuracy compared to BP.
>
> - The experiments are also rather limited: it's only two very similar datasets, and only shallow architectures. Convnets are also not used for cifar10 -- a few conv layers trained with BP can perform much better than ~56% with small MLPs, so seeing how NP-D performs there would be helpful.
>
> 6 and 9 layer experiments have been added to investigate NP-D’s scaling properties. Also, decorrelated BP was added as a baseline.
>
> - Why is everything trained for so many epochs? 2000 epochs is a lot, and backprop should normally converge to very good performance on cifar10 within the first few dozen epochs. But here it takes hundreds.
>
> The large number of epochs required has to do with the simple network architectures and the use of MSE loss instead of the more commonly used CCE loss. This was done because we wanted to compare NP, NP-D and BP in a straightforward setup, without worrying about complex interactions between architectural details, loss functions and the learning algorithms.

---

### Official Review · Reviewer_W31p · 2023-10-29

**Soundness:** 3 good
**Presentation:** 4 excellent
**Contribution:** 1 poor
**Rating:** 3
**Confidence:** 5

**Summary:**

This paper is about the node-perturbation approach to training artificial neural networks. The main claimed contributions of the paper are 1) reformulating the node perturbation (NP) methods in terms of directional derivatives (NP-iterative, section 2.1.2), and providing a simpler implementation (NP-activity, section 2.1.3) and 2) a decorrelation method (sec 2.2) applied on the input of each layer to speed up the convergence of training of node perturbation methods. The theory is then tested on classification tasks (CIFAR-10 and 100) with multi-layer perceptrons of three hidden layers (Sec 3.1, 3.2), and on convolutional networks (sec 3.3). An extension is also proposed based on two noisy forward passes instead of one clean and one perturbed pass (sec 3.4).

**Strengths:**

Strengths:

- **Clarity**. The paper is clearly written, and easy to follow. The concepts are well-explained, and the paper hierarchy is clear.

- **Ambition**. The paper is ambitious about the proposed method and test it only on more difficult benchmarks, CIFAR-10 etc, but it turns out to be also a weakness in this case (see weaknesses).

- **Importance**. The problem of finding alternatives to BP is important to reduce the energy cost of deep learning.

**Weaknesses:**

I see two main weaknesses:

- **Awareness of prior work**. The contribution of linking perturbations of forward passes to directional derivatives is not really a new contribution, it has been done before in e.g. [1], which I expected to see cited since it is extremely related. The activity-based updates can be thought of as a finite difference implementation of this approach. More generally, linking forward pass to directional derivatives means dealing with forward-mode differentiation, while BP is reverse-mode. Reverse mode is efficient because of the regime many-parameters/scalar loss, and forward mode is all the more inefficient if done exactly, hence the use of noise to probe multiple directions at once and make it more efficient, at the cost of more variance. It would be nice if the authors place their work clearly in this picture, and clarify their contributions after better literature review.


- **Overfitting**. The decorrelation approach proposed in this work does help for training accuracy, but leads to massive overfitting compared to BP (Fig 3 and 5). I think this is very concerning since achieving small training error is not the hard part of training neural networks, what really matters is the held-out error. It is known e.g. that BP with SGD has an implicit bias that enables the network to generalize [2], and somehow the decorrelation method suppresses this bias, which is interesting. I think this important point should be addressed/solved by this submission. Maybe the authors should try to vary the amount of decorrelation to see whether it affects overfitting. For now, the claim in the abstract that decorrelation makes NP learning competitive with BP is not backed by the data given the overfitting.

[1] Baydin, Atılım Güneş, et al. "Gradients without backpropagation." arXiv preprint arXiv:2202.08587 (2022).

[2] Chizat, Lenaic, and Francis Bach. "Implicit bias of gradient descent for wide two-layer neural networks trained with the logistic loss." Conference on Learning Theory. PMLR, 2020.

**Questions:**

Questions:

- How do the angles reported in Fig 2 evolve during learning? Do they become more aligned or less aligned?

- Why use the squared error loss when doing classification? Why not the cross-entropy loss? I can understand why the authors wanted to only use CIFAR-10/100 because these tasks are more difficult than MNIST or Fashion MNIST, but in this case it could be valuable to add experiments on MNIST to see whether the findings are the same regarding overfitting.

---

> ### Author Response · Authors · 2023-11-23
>
> We thank the reviewer for their thoughtful review and commentary.
>
> - Awareness of prior work.
>
> In the introduction, a mention has been added of Baydin et al., 2022; Ren et al., 2022, who do link NP to directional derivatives. We also note that our work is quite different from theirs, because in their work noise is used to estimate gradients, but their networks aren’t set up as noisy systems that propagate noise from previous layers to later layers.
>
> - Overfitting
>
> It is true that decorrelation increases overfitting, which we acknowledge in the text and in the discussion. We would like to point out, however, that despite the overfitting, test performance for NP-D is still much better than for NP.
>
> - How do the angles reported in Fig 2 evolve during learning? Do they become more aligned or less aligned?
> We did not measure the alignment across training, though we would expect that the measured angle ranking would not change, simply their magnitude would be affected by correlations in the network weights.
>
> Why use the squared error loss when doing classification? Why not the cross-entropy loss? I can understand why the authors wanted to only use CIFAR-10/100 because these tasks are more difficult than MNIST or Fashion MNIST, but in this case it could be valuable to add experiments on MNIST to see whether the findings are the same regarding overfitting.
>
> MNIST was initially used for some experimentation, but we found that, because of its simplicity, accuracy quickly saturated close to 100%, making it hard to distinguish the algorithms in terms of performance, especially visually. CIFAR-10 seems to be hard enough to make low-performing algorithms fail, but still lightweight enough to allow rapid development and iterations and the use of relatively simple networks.
>
> MSE was used instead of CCE mostly because of its simplicity and the stability of it’s loss magnitude, though none of the theory is different for alternative loss formulations.

---

### Official Review · Reviewer_G9as · 2023-11-02

**Soundness:** 3 good
**Presentation:** 3 good
**Contribution:** 2 fair
**Rating:** 3
**Confidence:** 4

**Summary:**

The paper studies node perturbation (NP), i.e. a category of learning algorithms that are different from BP, potentially more biologically-plausible, and possibly more suitable for efficient learning hardware, especially for noisy electronics. The work here formulates NP into a version that approximates directional derivatives with respect to a layer's activation, and also combines it with a method that decorrelates the activity among the neurons of a layer, resulting in faster and more stable learning.

**Strengths:**

The work is in an interesting field, namely that of alternative learning algorithms to backpropagation.

The paper is quite nicely written, making the reasoning for the design choices easy to follow.

The results seem potentially very useful to NP, stabilising and speeding up the learning process.

**Weaknesses:**

The work is only relevant to the niche of node perturbation, as the paper fails to contextualize the work more broadly, with literature review, experimental comparisons or otherwise.

The paper does not cite the state of the art (SOTA) within the NP subfield (Mengye Ren et al., ICLR 2023).

The methods that were used in that NP SOTA have not been incorporated in the experiments here, so it is hard to evaluate whether the advantages seen here could be combined with the previously seen progress in NP.

The authors do not cite any of the multiple other bio-plausible alternatives to BP than NP. The SOTA in such alternatives actually outperforms NP and is arguably more plausible and suitable for hardware (Journé et al., ICLR 2023).

The test accuracies reported here are significantly lower than that SOTA, and tests in more advanced datasets than CIFAR 100 have not been performed.

The paper's abstract concludes "making it competitive with BP", but this seems heavily exaggerated based on the actual demonstrations.

**Questions:**

Could the authors comment on the efficiency and the biological plausibility of the decorrelation method?

Similarly, how could the other aspects of the method be implemented in the brain?

For example, how could the "clean" (sic) version of the output be obtained in the brain, as it assumes the absence of noise?

Does the claim of the paper about suitability of noisy hardware hold, under the assumption of a clean forward pass?

How plausible is it biologically that for each training example, multiple forward iterations are needed before the full update? It seems that hundreds of such iterations were necessary in the simulation.

How energy efficient would this be in a hardware implementation? It seems to me that it would also increase the energy consumption by orders of magnitude.

The paper mentions that the convolutional architecture was very shallow. Why couldn't it be scaled up?

I could not find an experiment showing the impact of the decorrelation method on the results. Was it implemented in all versions of NP experiments?
Also, was it (or any other decorrelation method) used in experiments with backprop? Otherwise, could the authors comment on whether comparisons are fair?

---

> ### Author Response · Authors · 2023-11-23
>
> We thank the reviewer for their thoughtful review and commentary.
>
> - The work is only relevant to the niche of node perturbation, as the paper fails to contextualize the work more broadly, with literature review, experimental comparisons or otherwise.
>
> - The paper does not cite the state of the art (SOTA) within the NP subfield (Mengye Ren et al., ICLR 2023).
> The methods that were used in that NP SOTA have not been incorporated in the experiments here, so it is hard to evaluate whether the advantages seen here could be combined with the previously seen progress in NP.
>
> We added a brief mention of Baydin et al., 2022; Ren et al., 2022 to the introduction section, as those authors did indeed relate NP to directional derivatives. However, we also note that their forward gradient method is quite different from ours, as the perturbations in their work are only used to estimate gradients and no noise is actually propagated through the network. This would make their method less suitable for noisy systems, where the noise cannot be avoided. Our method focuses more on applications in noisy systems and even works without any clean forward passes, as shown in our “double noise” experiment. In addition, our ANP method requires no access to the noise vector, but simply measures performance differences and relates them to activity differences, helped by decorrelation of said activities. This method would presumably also work in actual noisy systems, where one does not know which part of an activation is signal and which part is noise.
>
> - The authors do not cite any of the multiple other bio-plausible alternatives to BP than NP. The SOTA in such alternatives actually outperforms NP and is arguably more plausible and suitable for hardware (Journé et al., ICLR 2023).
>
> There are indeed many alternative methods of bio plausible learning, like Feedback Alignment, Target Propagation, energy based methods and many others. The decision to focus on NP stems from our interest in noisy systems and their characteristics.
>
> - The paper's abstract concludes "making it competitive with BP", but this seems heavily exaggerated based on the actual demonstrations.
>
> The claims about being competitive with BP have been toned down a bit in the introduction and discussion sections, as our method does suffer from more overfitting.
>
> - The test accuracies reported here are significantly lower than that SOTA, and tests in more advanced datasets than CIFAR 100 have not been performed.
>
> The purpose of our study was not to achieve SOTA results but to do a very straightforward comparison between BP and NP, uncomplicated by performance enhancing engineering solutions like dropout, batch norm, momentum optimizers etc, which might confound the results with their many interactions.
>
> - Could the authors comment on the efficiency and the biological plausibility of the decorrelation method?
>
> A discussion of the plausibility of decorrelation has been added to the discussion section, pointing to lateral inhibition as a possibly analogous process in the brain.
>
> - Similarly, how could the other aspects of the method be implemented in the brain?
> For example,...?
>
>
> In terms of biological plausibility, the ANP method is the most straightforward to envision working in the brain, as it measures only activation differences and how they relate to a global reward signal, meaning no access to a noise vector is needed. Instead of the double forward pass implemented in our code, one could imagine how random fluctuations in neuronal activity are correlated by the brain to a global reward signal in a dynamical way, without clearly distinguishing between “passes”.
>
> - How plausible is it biologically... magnitude.
>
> These are fair criticisms, but they only apply to the NP-iterative method. The NP-iterative method was included as a “best case” example for decorrelated NP, but the NP-activity method is the one that is most biologically plausible and potentially cheap in terms of computation.
>
> - The paper mentions that the convolutional architecture was very shallow. Why couldn't it be scaled up?
>
> As CIFAR-100 isn’t a large dataset, we didn’t feel like a much larger convolutional architecture would be required, especially given our aim of making an uncomplicated comparison between decorrelated NP and BP. Scaling the methods described to datasets like ImageNet is intended as the subject of an extension of this work.
>
> - I could not find... comparisons are fair?
>
> Initially, we did not include decorrelated BP, but as of the current revision, we have included it as an additional baseline. We used the iterative decorrelation method from Ahmad et. al., 2022 as it’s a fast and provably reduces a specific loss function measuring the level of correlation in each layer. Alternative decorrelation could work but the specific method for decorrelation was not an interest point for this study.

---

### Official Review · Reviewer_hNrk · 2023-11-03

**Soundness:** 3 good
**Presentation:** 3 good
**Contribution:** 2 fair
**Rating:** 5
**Confidence:** 5

**Summary:**

The authors present an approach to improve the efficiency and stability of node perturbation (NP) as an alternative, biologically plausible method to backpropagation (BP) for credit assignment in deep neural networks. Their main contributions include reframing NP in terms of computing directional derivatives and proposing a decorrelation procedure to mitigate input bias. Specifically, they suggest using a decorrelation matrix initialized as the identity matrix and subsequently updated to reduce correlations. For gradient approximation, they perturb one layer at a time using a noise vector, measuring the resultant change in loss, and thereby estimating the gradient respective to each layer.

**Strengths:**

**Novelty in Approach**: The manuscript's primary strength lies in its enhancement to NP by directional derivatives calculated from neural activity measurements. This offers a fresh lens to potentially harness NP more effectively in deep learning contexts -- perhaps also spur methods utilizing a hybrid NP_noise and NP_activity update.

**Clarity in Gradient Approximation**: The iterative node perturbation method introduced offers a transparent and logical pathway to approximate gradients. By perturbing one layer at a time and measuring the resultant loss changes, the approach provides both intuition and efficacy in gradient estimation.

**Weaknesses:**

**Learning Rate Optimization**: The approach to learning rate optimization might introduce inadvertent biases. By optimizing the learning rate specifically for NP and then using this optimized rate for BP without independent optimization, the results might not reflect the accurate comparison with BP. This is further underscored by references like Hiratani et al., which hint at nuances between NP and BP at different learning rates.

**Concerns Over Biological Plausibility**: The introduction of the decorrelation mechanism, while effective, does raise eyebrows regarding its biological realism. If the overarching goal of the research is to be in harmony with biologically plausible learning paradigms, then not sure how the decorrelation step fits in.

**Ambiguous Visual Interpretation**: Interpretation of Figure 2 seemed ambiguous to me. With training curves of three methods appearing quite similar without decorrelations, the clarity of conclusions derived becomes muddied.

**Scope Limitation with Network Types**: The research seems to overlook certain types of networks, notably the absence of comparisons for NP iterative in convolutional networks on established benchmarks like CIFAR100.

**Formatting and Implementation Details**: The paper contains formatting discrepancies, notably in equation numbering. Such inconsistencies can be distracting and hinder smooth comprehension. Additionally, the omission of granular details about the implementation, such as the underlying framework, time metrics, and codebase accessibility.

**Questions:**

1. Do the authors plan to release the code base? If so, could you please share the link to an anonymized repository?

2. What exactly does figure 2 convey? The authors say “ranking” is important here – for layer 2 NP iterative is slightly better, whereas for output NP activity is slightly better. Overall it’s difficult to see what should be the conclusion since the training curves for these 3 methods without decorrelations is very similar.

3. Grid search for learning rate optimization – lr is optimized for NP, and then BP is trained with this lr. Hiratani et al. have shown that NP approximates BP in low lr – for a fair comparison of the learning curves, BP curve should be for LR optimal for that? Also, the variance in performance with multiple seeds, across learning rates will be good to see. How does this scale with the network architectural parameters, such as the depth and width of a fully connected network?

4. Hiratani et al. have also shown that NP is unstable in higher learning rates, did you observe “crashes” in training – where the accuracy suddenly falls to chance? It would be interesting to see whether decorrelations help with this.

---

> ### Author Response · Authors · 2023-11-23
>
> We thank the reviewer for their thoughtful review and commentary.
>
> - Learning Rate Optimization
>
> The reviewer appears to be mistaken - we optimized learning rates separately for every algorithm and experiment to avoid biasing our results. We explain this more clearly in the methods section and appendix in our updated revision.
>
> - Concerns Over Biological Plausibility
>
> The decorrelation step as currently implemented is indeed not exactly biologically realistic. We would maintain, however, that something akin to decorrelation does happen in human brains, possibly through processes like lateral inhibition. We added this point to the discussion section along side relevant literature references.
>
> - Ambiguous Visual Interpretation
>
>  It is true that the ANP method doesn’t outperform NP nearly as much as does INP. This is now more clearly expressed in the text and conclusions. The reason for emphasizing the activity based method is that it does not require access to the noise generation process, making it more bio-plausible than the iterative method, while also making it potentially more suitable for certain kinds of hardware.
>
> - Scope Limitation with Network Types
>
> We have now added the DINP method to the CIFAR-100 convnet experiments. We also included 6 and 9 layer networks on CIFAR-10.
> Formatting and Implementation Details More implementation details have now been added. The codebase itself will be released upon acceptance.
>
> - Do the authors plan to release the code base? If so, could you please share the link to an anonymized repository?
>
> Yes, the codebase will be released.
>
> - What exactly does figure 2 convey? The authors say “ranking” is important here – for layer 2 NP iterative is slightly better, whereas for output NP activity is slightly better. Overall it’s difficult to see what should be the conclusion since the training curves for these 3 methods without decorrelations is very similar.
>
> The main message in Figure 2 is that the standard formulation of NP (noise method) can be aligned more closely with BP by the activity based method and even more so with the iterative method, though for the latter method this comes at the expense of additional computation. Based on our theoretical work, the iterative method is expected to align more closely to BP than the activity method in all layers. We suspect the lack of outperformance in the final layer might be a statistical issue, as that layer contains only 10 neurons, making the measurement process very noisy.
>
> - Grid search for learning rate optimization – lr is optimized for NP, and then BP is trained with this lr. Hiratani et al. have shown that NP approximates BP in low lr – for a fair comparison of the learning curves, BP curve should be for LR optimal for that? Also, the variance in performance with multiple seeds, across learning rates will be good to see. How does this scale with the network architectural parameters, such as the depth and width of a fully connected network?
>
> The LR was optimized separately for each algorithm, precisely to avoid this bias. This will be explained more clearly in the text. As for the variance between seeds, we show not only the mean (thick line), but also the best and worst seeds (shaded areas) to give an impression of the variance. Because the variance was fairly low in most experiments, this is not easy to spot in the graphs. A scaling experiment has been added where 6 and 9 hidden layers were explored in addition to 3 hidden layers.
>
> - Hiratani et al. have also shown that NP is unstable in higher learning rates, did you observe “crashes” in training – where the accuracy suddenly falls to chance? It would be interesting to see whether decorrelations help with this.
>
> Yes, we found that there would be a LR where NP learning would crash almost immediately, then there would be a slightly lower LR where the algorithms would crash after a few hundred epochs and finally there would be the highest stable LR, which we used for our experiments.
> The exception here is the iterative NP method. For this method, if training was stable initially, it would remain so throughout training, as this method does not suffer from exploding weights the same way as other NP formulations.

---

### Author Response · Authors · 2023-11-23

We thank all reviewers for their interest in our research and for their effort and constructive criticisms. We have made a fairly major revision to the paper including the following improvements:

- Prior work regarding perturbation learning and directional derivatives was cited in the introduction, along with an explanation of how our method differs. (Baydin et al., 2022; Ren et al., 2022 )
- The contributions made in our work are now explained more clearly in the introduction, especially regarding the biological plausibility advantages of activity based node perturbation (ANP).
- Claims about our method being "competitive with BP" were toned down a bit and the contextual nature of our findings was emphasized more.
- Claims regarding biological plausibility are now explained more in the discussion.
- All experiments were run with 5 instead of 3 random seeds.
- Scaling experiments with 6 and 9 hidden layers were added for CIFAR-10.
- Decorrelated BP was added as baseline to most experiments.
- Several improvements to notation, consistency of terminology etc. were made.
- Performance curves of algorithms are colored consistenly across figures and experiments.

---

### Meta-Review · Area_Chair_xXYz · 2023-12-07

**Metareview:**

This paper introduces various new tricks to make Node Perturbation, a biologically-plausible alternative to backpropagation, perform better. These tricks include using directional derivatives and decorrelating each layer's input activities. The reviewers raised concerns most importantly about the biological plausibility of the approach and benchmarking against existing methods.

**Justification For Why Not Higher Score:**

The AC recommends rejection based on the concerns brought up by reviewers.

**Justification For Why Not Lower Score:**

N/A

---

### Decision · Program_Chairs · 2024-01-16

Reject